# Learning cellular morphology with neural networks

Philipp J. Schubert [1], Sven Dorkenwald[1], Michał Januszewski [2], Viren Jain[3] & Joergen Kornfeld [1]

Reconstruction and annotation of volume electron microscopy data sets of brain tissue is challenging but can reveal invaluable information about neuronal circuits. Significant progress has recently been made in automated neuron reconstruction as well as automated detection of synapses. However, methods for automating the morphological analysis of nanometer-resolution reconstructions are less established, despite the diversity of possible applications. Here, we introduce cellular morphology neural networks (CMNs), based on multi-view projections sampled from automatically reconstructed cellular fragments of arbitrary size and shape. Using unsupervised training, we infer morphology embeddings (Neuron2vec) of neuron reconstructions and train CMNs to identify glia cells in a supervised classification paradigm, which are then used to resolve neuron reconstruction errors. Finally, we demonstrate that CMNs can be used to identify subcellular compartments and the cell types of neuron reconstructions.

[1] Max Planck Institute of Neurobiology, Electrons - Photons - Neurons, 82152 Planegg-Martinsried, Germany. [2] Google AI, Zurich 8002, Switzerland. [3] Google AI, Mountain View 94043 CA, USA. Correspondence and requests for materials should be addressed to P.J.S. (email: pschubert@neuro.mpg.de) or to J.K. (email: kornfeld@neuro.mpg.de)

Advances in volume electron microscopy (VEM) have led to increasingly large three-dimensional (3D) images of brain tissue, making manual analysis infeasible[1]. Multi-beam scanning electron microscopes[2] and transmission electron microscopes equipped with fast camera arrays can now generate data sets exceeding 100 TB[3], a development that was fortunately accompanied by substantial progress in neuron reconstruction[4–9] and the automatic analysis of synapses[10–13]. These advances have enabled automatic morphology analyses on the neuron (fragment) scale, which were previously restricted to direct segmentation error detection[5,14] or the use of manual skeletons with data-specific hand-crafted features[11,15,16]. Cell types and other biological properties that can be inferred from the morphology of a neuron are required for the interpretation of a connectome[11] and can also constrain automatic neuron reconstruction itself[17].

Outside of the field of connectomics, many approaches have been developed for automated 3D shape analysis, including multi-view two-dimensional (2D) projection-based neural network models[18,19] as well as voxel- and point cloud-occupancy-based 3D networks[20,21]. Interestingly, projection-based models often appear to outperform 3D architectures, possibly because of higher-resolution input due to decreased model complexity[18].

Here we present cellular morphology neural networks (CMNs), which use multi-view projections to enable the supervised and unsupervised analysis of cell fragments of arbitrary size while retaining high resolution. First, we demonstrate that CMNs can be used to automate morphology feature extraction itself by inferring low-dimensional embeddings, dubbed Neuron2vec, through unsupervised triplet-loss training[22,23]. Second, we apply CMNs to the supervised classification of glia cells and use these data to demonstrate the feasibility and effectiveness of a simple top–down false merger resolution strategy. Third, we identify neuronal cell types and compartments, outperforming methods with hand-crafted features based on skeleton representations on the same data, and finally perform high-resolution cell surface segmentation to identify dendritic spines.

## Results

### Cellular morphology learning networks

CMNs are convolutional neural networks (CNNs) optimized for the analysis of multi-channel 2D projections of cell reconstructions, inspired by multi-view CNNs for the classification of objects fitting into projections from one rendering site[18,19].

Su et al.[18] rendered views of an entire object, potentially sacrificing crucial detail when applied to reconstructions of entire neurons (or very large objects in general), which can have processes as thin as 50 nm extending over millimeters[24]. To address this problem, we used a sampling algorithm that homogeneously probes an entire cell at many locations ("Methods"; Fig. 1a, b) that are then analyzed either individually or in combination by a CMN. The neuron reconstructions were taken from a songbird basal ganglia data set and consisted of flood-filling network (FFN)-created supervoxels[25] (SVs), which were agglomerated to sets of super-SVs (SSVs)[8], each corresponding to a single neuron. A rectangular field of view (FoV) was chosen for the projection to capture the elongated shape of most neuronal arbors more effectively. Additionally, we incorporated image channels beyond the cell shape (here represented through depth-map projections) of the same rendering perspective. This extension allows the CMN to analyze the geometry and density of objects contained in a cell, such as mitochondria and other organelles (Fig. 1c, d).

We implemented the view rendering engine with OpenGL and the neural network models using the ElektroNN neural network library (www.elektronn.org) and adapted the SyConn pipeline (https://github.com/StructuralNeurobiologyLab/SyConn/; see Supplementary Table 1 for a timing overview). The models were trained using standard loss optimization procedures (Adam or stochastic gradient descent) on various supervised and unsupervised tasks (see Fig. 1e for an application overview), which are described in the following to demonstrate the versatility of the approach.

### Neuron2vec embedding

Supervised models often require hard-to-obtain manual ground truth, making alternative objective functions and models based on intuitive isomorphisms of the underlying data desirable. We trained a CMN using triplet loss[23] to learn a latent space (embedding; dimension $d_z = 10$) of single renderings based on the similarity of three inputs $x_{\text{ref}}$, $x_+$, and $x_-$. At every location, two renderings (see above; $\varphi = 50°$) were generated that served as a similar pair ($x_{\text{ref}}$ and $x_+$). In contrast, a single rendering at a different, randomly sampled location was used as the dissimilar example $x_-$ ("Methods"). During training, the views were sampled from 372 cell or cell fragment reconstructions (181.02 mm; 19.95 giga voxel (GV); 32,324 μm³).

We explored the information content of the embedding through inspection of clusters in a 2D t-SNE[26] projection (Fig. 2a) of an example cell reconstruction (Fig. 2b; colors as in Fig. 2a; see "Methods") and by fitting a $k$-nearest neighbor classifier (kNN; $k = 5$; uniform weights) to the Neuron2vec encoding extracted from a set of neurites with cellular compartment ground truth annotations (same as in "Cellular compartment identification"; total path length: 30.16 mm; 3.05 GV; 4947 μm³).

The predictive performance of the kNN classifier on the triplet-CMN (t-CMN) latent vectors of a cellular compartment test set (same as in "Cellular compartment identification"; 20.75 mm; 1.31 GV; 2130 μm³) was particularly low for dendrites (F1-score for dendrite: 0.565, axon: 0.686, soma: 0.942; Supplementary Fig. 1). We were able to increase the classification performance by sampling the similar view $x_+$ from close-by rendering locations during training as well (two and eight additional rendering locations $N_r = 3$ and 9), which had a smoothing effect across the inferred latent vectors of adjacent rendering locations (Supplementary Fig. 2; F1-score for dendrite, axon, soma with $N_r = 3$: 0.751, 0.767, 0.951; $N_r = 9$: 0.804, 0.800, 0.951). Increasing the latent space dimension to $d_z = 25$ had little impact on the model's performance (F1-score for dendrite, axon, soma with $N_r = 3$: 0.728, 0.756, 0.957; $N_r = 9$: 0.797, 0.808, 0.967).

Axons, or thin processes in general, and soma regions could be readily identified based on a color projection of the embedding using principal component analysis (PCA; Fig. 2 and Supplementary Fig. 2), whereas, for example, spiny dendrites showed a heterogeneous and less continuous spectrum (inset Fig. 2b). Interestingly, views did not only group by compartment type but also formed sub-clusters separating views with cell objects (such as mitochondria and synaptic junctions) from those without (first and second column in Fig. 2c), providing an explanation for the low kNN performance.

We next took a supervised end-to-end approach and examined ground truth-based CMN models.

### Glia detection and top–down segmentation error correction

Owing to the dense heavy-metal staining used for VEM data, the acquired images contain all cell types, including astrocytes and other glia types, which are usually not considered for connectomic analysis. Astrocytes have tight surface contacts with neurons to supply them with nutrients and regulate the local environment[27], making astrocyte–neurite mergers a problem in automated reconstruction pipelines. While it can be difficult for humans to determine from the raw EM data whether a process

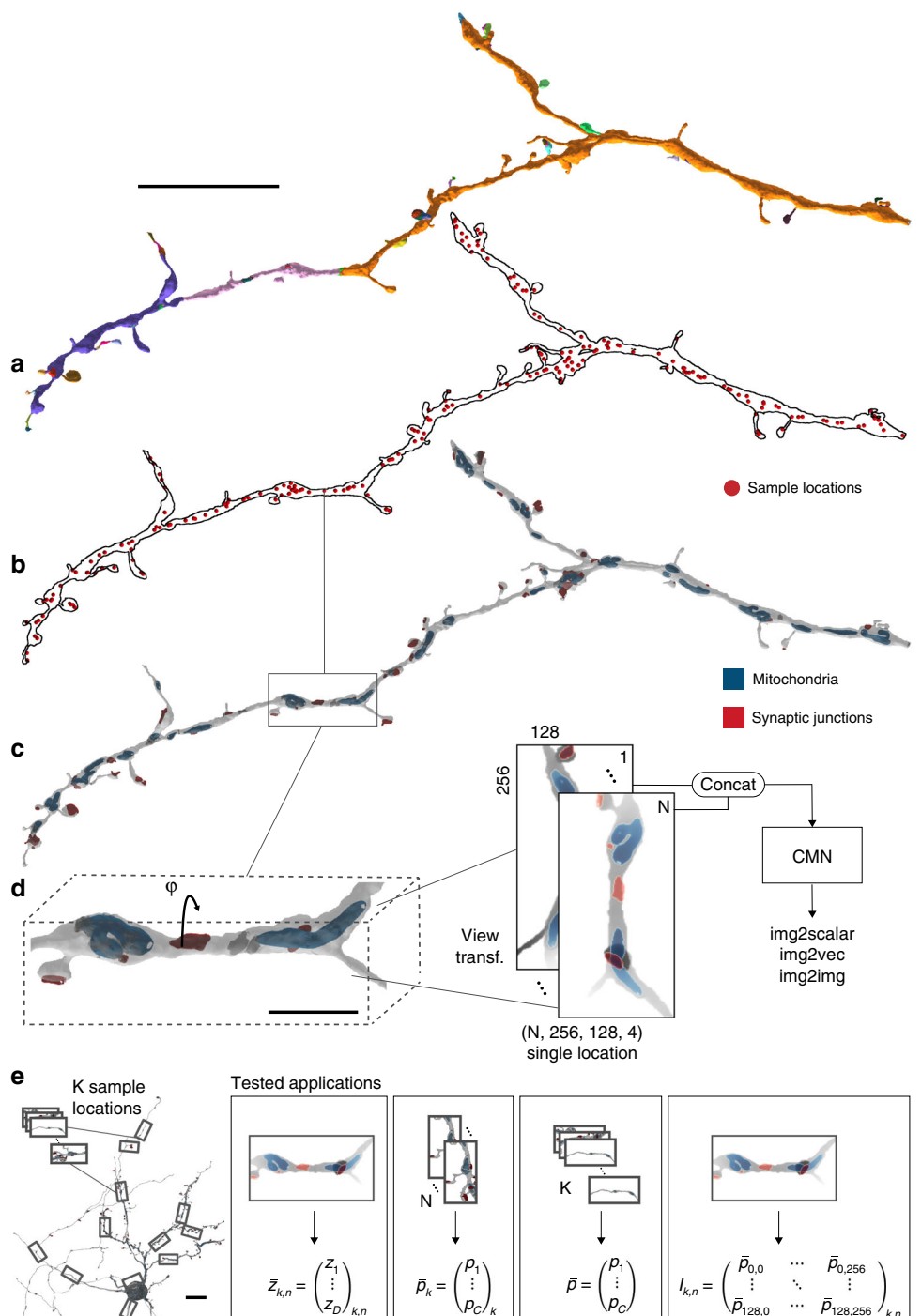

**Fig. 1** Multi-view generation for a given neurite reconstruction and model architecture. **a** Flood-filling network reconstruction and agglomeration of a dendrite, with supervoxels individually colored. **b** Sampled multi-view projection locations indicated as red spheres. **c** Two-dimensional projection of the whole neurite reconstruction with cell organelles (blue: mitochondria; red: synaptic junctions). **d** Each location was rendered from two different perspectives, one orthogonal to the first and second principal component (p.c.), the other one rotated by $\varphi$ around the first p.c., i.e., orthogonal to the first and third p.c. (here: $\varphi = 90°$). **e** Multi-view fingerprints were extracted from the neuron reconstruction and served as input for one unsupervised and three supervised convolutional neural network-based applications, from left to right: Neuron2vec latent vector inference from single views to encode morphology; (multi-) class probability from multi-views to infer cellular compartments and glia fragments; multi-class prediction of many multi-views (*K*-views) for morphological cell-type identification; high-resolution semantic segmentation on single views to identify individual spines. Scale bars are 10 µm in **a**, **e** and 2 µm in **d**

belongs to a glia cell, it seemed straightforward to solve this classification task using the established multi-view representation due to their distinct shape (Fig. 3a, Supplementary Fig. 3). A CMN was trained and validated on a set of manually annotated SVs (from 34 neurite reconstructions and 118 glia SVs;

368.74 mm; 16.48 GV; 26,695 µm³), achieving an average F1-score of 0.979 (precision: 0.985; recall: 0.974; *N*: 9695 multi-views; Supplementary Fig. 4). We then explored the effect of varying data context and view resolution on classification performance and found that, as expected, context is crucial (largest context

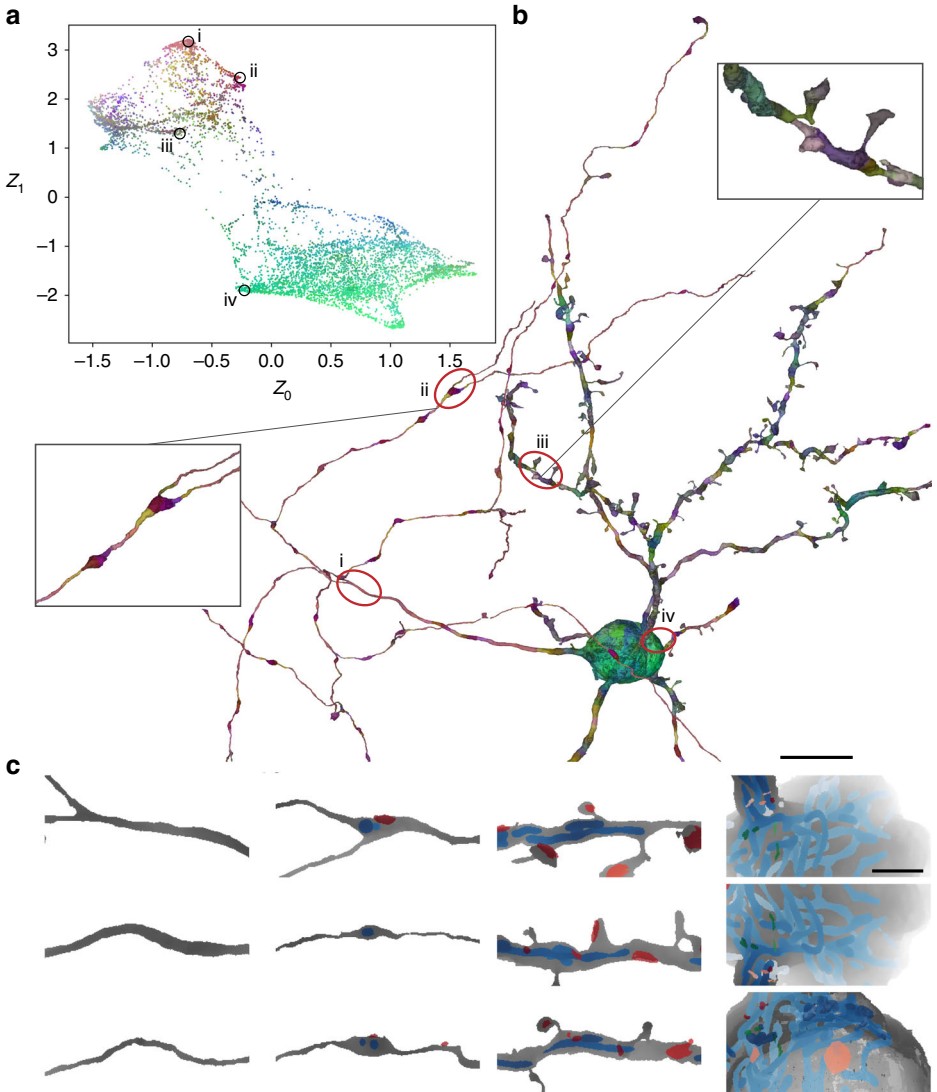

**Fig. 2** Neuron2vec embeddings learned by a triplet-convolutional neural network ($d_z = 10$; $N_r = 1$). **a** t-SNE-transformed latent space (top three p.c.s covered 60.2% of the variance and were used as RGB values) of an example cell reconstruction, shown in **b** (coloring as in **a**). Insets show close-ups of an axonal bouton and a spiny dendrite. **c** The three nearest neighbor views of locations i–iv in **a** and **b** from left to right column: axonal segments, axonal segments with mitochondria (blue) or synaptic junctions (red), spiny dendrite segments, and at the soma (green: vesicle clouds). Scale bars in **b** and **c** are 10 and 2 μm, respectively

tested: $8 \times 4 \times 4\ \mu m^3$; F1-score of 0.900 upon 75% reduction; Supplementary Fig. 4), while reduced resolution barely affects the performance of CMNs (F1-score of 0.967 upon 75% reduction). In agreement with the validation results, we evaluated a set of test SVs ($N_{neuron}$: 84; $N_{glia}$: 85; 27 μm$^3$) and found that SVs belonging to larger automatic neurite reconstructions (SSV bounding box diagonal (BBD) >8 μm) showed a significantly better performance (F1-score 0.964) in comparison to the set of all SVs, which included many small and hard to classify fragments (F1-score 0.868; see Fig. 3b first histogram bin).

Can we exploit the excellent glia classification results to reduce the rate of false mergers in a segmentation? The naive solution would be the simple removal of all SVs classified as glia fragments, but this approach could induce large-scale false splits in the case of false positive classifications (Supplementary Fig. 6a). We therefore developed a more sophisticated splitting heuristic, ensuring that large neuron fragments remained connected in the presence of small misclassifications.

The agglomeration of SVs can be represented as a graph, connecting adjacent SVs (nodes) of the same SSV (neurite) with edges. In this graph, glia predictions were used as node properties to identify connected glia and neuron components with their respective sizes. Sufficiently large (BBD ≥8.0 μm) glia components were removed and added to the glia graph, while small glia components (BBD <8.0 μm) remained in the original SSV graph. The resulting connected components of the graph were stored as individual glia and neuron SSVs, respectively (Fig. 3c, d; see "Methods").

We first evaluated the splitting procedure by assessing how many SVs remained wrongfully assigned and manually labeled the SVs of 12 neurite reconstructions as ground truth ($N_{SV}$: 616; 5161 μm$^3$). As before, our approach showed a significantly higher performance when classifying large SVs and after the splitting procedure, almost the entire glia share of the original SSVs was removed (average F1-score with volume weights: 0.995, neuron: 0.997, glia: 0.981; unweighted, i.e., SV-level: 0.937, neuron: 0.939, glia: 0.934; support for $N_{neuron}$: 321 SV, 4534 μm$^3$; $N_{glia}$: 295 SV, 628 μm$^3$; Fig. 3e, Supplementary Fig. 6b, c). In the entire data set, 4,324,836 SVs were classified as neuron (606,922 μm$^3$) and 1,696,173 as glia (110,342 μm3), yielding an astrocyte volume

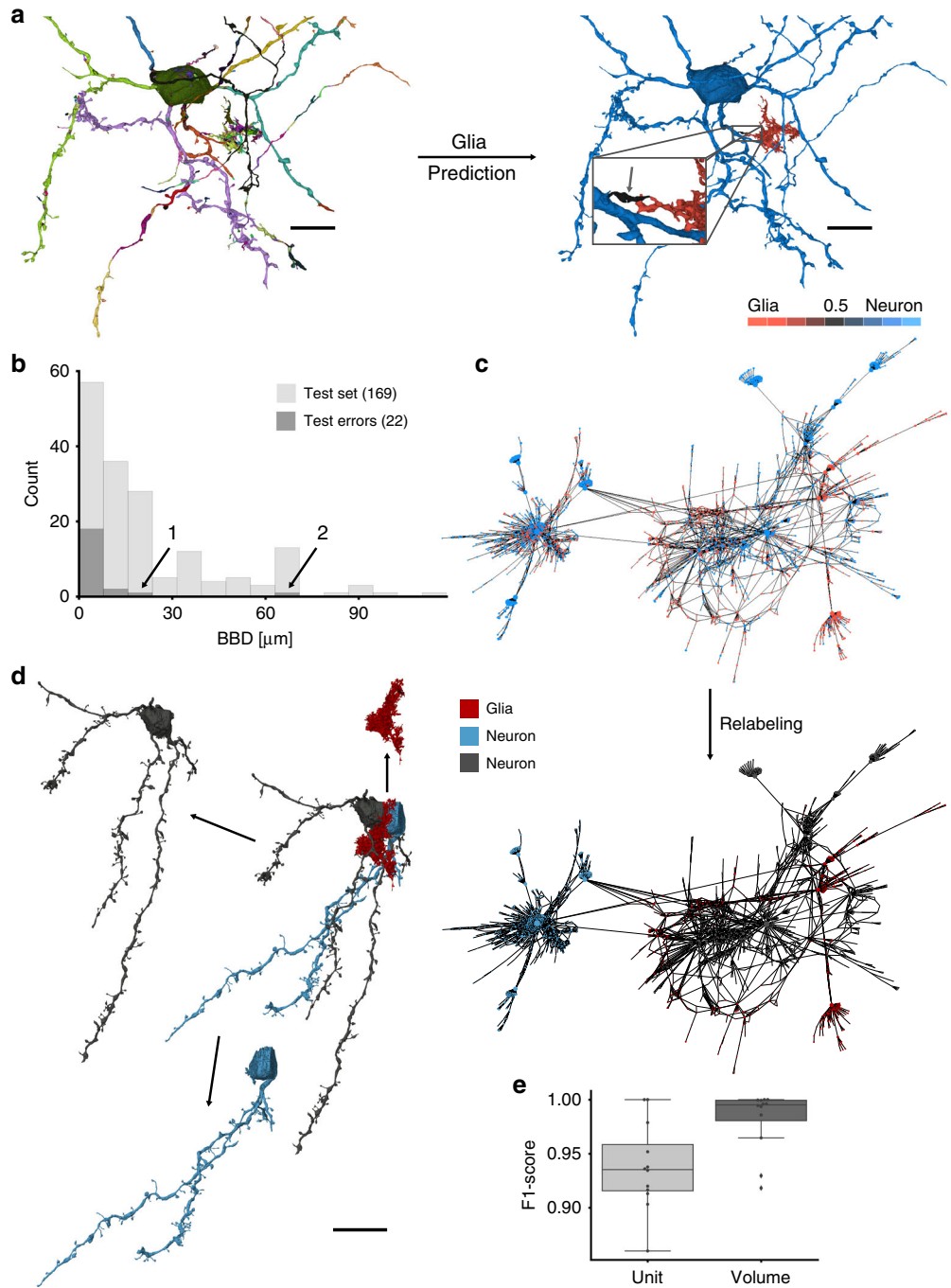

**Fig. 3** Glia prediction and reconstruction splitting procedure. **a** Each supervoxel (SV) (unique coloring) of a neurite reconstruction was classified as glia (red) or neuron (blue) based on the cellular morphology neural networks output. **b** Corresponding neurite length (bounding box diagonals (BBDs)) distribution for 169 randomly drawn SVs (light gray) vs. SVs with classification errors (dark gray). Arrows indicate errors of SVs with a BBD ≥10 μm (see Supplementary Fig. 5 for example renderings). **c** Super-SV (SSV) graph representation (spring-layout) of a glia-predicted neurite reconstruction (SV volume represented by node size), compared with Supplementary Fig. 6a. **d** Connected component (CC) meshes (left) and SSV graph (right) indicating CCs after the splitting procedure. **e** Boxplot (box: median, first and third quartile (Q1 and Q3); whiskers extend to first and last data point within 1.5 times the interquartile range (Q1–Q3) below Q1 and above Q3, respectively) of SV classification performance with unit weights (light gray) and volume weighted (dark gray) splitting performance of 12 neurites (*p* value of Wilcoxon–Mann–Whitney two-sample rank-sum test: *p* = 0.028 < *α* = 0.05 with *N* = 12). Scale bars are 10 μm for **a**, **b** and 20 μm for **d**

fraction of about 0.154. It should be noted that the EM data set was prepared so as to preserve the extracellular space, which may change the morphology of the glial processes[28].

While this evaluation already showed that the splitting heuristic successfully removes large volume fractions of glia from neurite SSVs, it is necessary to assess whether it could do so

without introducing too many new false splits. Note that the underlying FFN segmentation has already an exceptionally low false merger rate, which was evaluated in depth in Januszewski et al.[8]. Consistent with this, the splitting procedure created only 882 additional SSVs from the 181 affected SSVs in the entire data set. We therefore inspected 180 SSVs (one very large SSV with

600,924 SVs was evaluated separately by sampling, see "Methods") and their resulting connected components (470 SSVs) to estimate how many false mergers were resolved at the cost of introducing new false splits into neuron SSVs, which would require manual fixing. About half (55%; 122 out of 222 neuron SSVs) of the newly created neurite SSVs were affected by a new false split, which severed them off of the originally correct (regarding false splits) neurite. In 7% (16 out of 222 neuron SSVs), the procedure removed only a small neuron component (e.g., parts of the soma–internal Golgi falsely classified as glia fragment). The other 38% (84) of neurite SSVs were correctly recovered meaning that all their false mergers with glia, or with other neurons bridged through a glia cell, were removed without the introduction of a new false split (see Fig. 3d and Supplementary Fig. 6f for examples). These numbers were comparable on the sampled SSVs, which originated from the very large SSV (58% affected by new false splits, 42% correct). While we could not accurately measure the remaining neuron–glia false merger rate after splitting due to the effort involved in inspecting a large part of the data set to estimate the rate of these rare events (inspection of a random sample of 100 SSVs revealed no glia merge errors), any remaining glia merge error is expected to be small (volume F1-score 0.995, Fig. 3e).

We then attempted to reconstruct the 27 astrocytes identified in the data set starting at their somata and assigning each glial fragment to the closest astrocyte soma (see "Methods"), justified by reports that astrocytes establish roughly spherical, largely non-overlapping territories[29]. While the CMN-based glia identification appears promising in principle (see Supplementary Fig. 7 for automatically extracted glia–blood vessel contacts), our approach likely merged arbors from other glia cells because they reach into the data set from the outside.

**Neuron-type classification**. Similar to glia cells, many neuronal types can be identified based on their morphology[15], a feature that was used well before connectivity-based methods[30] and other approaches (reviewed in ref. [31]). We recently demonstrated on the same songbird data set that a feature-based method with random forest classifiers (RFCs) can be used to identify the four main cell types, excitatory axons (EA), medium spiny neurons (MSN), pallidal-like neurons (GP), and interneurons (INT)[11]. However, manual neurite skeletons and hand-designed feature vectors were required as basis for this classification method.

In contrast to the classification of astrocytes, neuronal-type classification is less successful when using only a local view (i.e., spatially focused) of a neurite (F1-scores for single views: 0.885 and after majority vote: 0.891; training set: 145.98 mm, 17.21 GV, 27,872 µm$^3$; test set: 65.04 mm, 7.31 GV, 11,843 µm$^3$). A simple solution would be to increase the FoV for a single view, capturing the entire extent of a neuron, similar to the proposed object classification method by Su et al.[18]. However, this approach reduces the resolution for a given view size, obscuring potentially important details. Alternatively, local views from different locations can be sampled at random and combined to a global representation (multi-views of size $N$, further called N-views; Fig. 4a) by the neural network model. This approach does not sacrifice resolution and increased the classification performance substantially ($N = 10$ views: F1-score of 0.987, and 0.970 on SSV; Fig. 4b). It should be noted that lower-resolution, zoomed out views might still be beneficial or view location information additionally fed into the network but we did not explore this further. Interestingly, a greater number of sample locations did not necessarily increase performance (F1-score for $N = 60$: 0.984, SSV: 0.957; Fig. 4b), possibly due to increasing model complexity and decreasing diversity of the training data through fewer

independent view samples per neuron. This result was consistent with the observation that the N-view F1-scores throughout all $N$s were significantly lower without shuffling (views were analyzed in the order they were created; Fig. 4c), which led to spatially correlated N-view sets, i.e., they likely consisted of views from a single compartment type only. For all $N$, additional models were trained to predict sets of views without cell organelle channels, which, in agreement with our previous results using RFCs[11], reduced the F1-score on SSV-level substantially (e.g., F1-score reduction of 0.088 for $N = 20$ with majority vote; Fig. 4b).

**Cellular compartment identification**. We next attempted to analyze the FFN neuron segmentation by identifying subcellular compartments (axon, dendrite, and soma) at single multi-view locations (Fig. 5a).

Similar to the glia model, a reduction of the original FoV of $8 \times 4 \times 4$ to $2 \times 1 \times 1$ µm$^3$ reduced the performance (F1-score on validation set of 0.996 with original views vs. 0.913), while a four-fold downsampling of the multi-views had almost no effect (reduction of 0.014, Fig. 5b). Not surprisingly, the performance of the soma class was barely affected by this or any other changes to the input, whereas the discrimination between axons and dendrites was strongly dependent on cell organelle information (F1-score reduction by 0.08 with exclusion; Fig. 5a, b).

The performance of the CMN approaches were directly compared with the skeleton-based RF classification developed by us previously[11] on a set of 28 manually annotated reconstructions (20.75 mm; 1.31 GV; 2130 µm$^3$). Two different FoVs (implemented as maximum skeleton traversal distances for the RFC approach, RFC-4, with 4 µm and RFC-8 with 8 µm) were tested with morphology features extracted from these FoVs (see "Methods") and fitted to the same training data as the CMN (path length 30.16 mm; 3.05 GV; 4947 µm$^3$). Although the CMN (2 views; Fig. 5b) was operating on much less context (multi-views were based on a $8 \times 4 \times 4$ µm$^3$ subvolume vs. a maximum possible subvolume of $16 \times 16 \times 16$ µm$^3$ for the RFC-8 model), it outperformed both skeleton-based models and the Neuron2vec-kNN classifier (t-CMN; Fig. 5c; average F1-score values for t-CMN: 0.834; CMN: 0.955; RFC-4: 0.761; RFC-8: 0.779). Owing to the low predictive performance of the skeleton-based RFC-4/8 models on the soma class (see Fig. 5c and Supplementary Fig. 8a for an example skeletonization; a sliding window majority vote could not account for the difference in performance, see "Methods" and Supplementary Fig. 8b, c), we performed an additional evaluation on test data of axons and dendrites only (RFC*-4, RFC*-8). The performance values indeed increased substantially (RFC*-4: 0.912; RFC*-8: 0.948), to almost comparable levels of the CMNs, at least on this narrower classification problem.

**High-resolution semantic segmentation of neurite surfaces**. The neuron classification described thus far is restricted in its spatial resolution to the minimum size of at least one view rendering ($8 \times 4 \times 4$ µm). This property makes the approach suitable for the identification of neuronal compartments on a larger scale (axons, dendrites, and somata), a situation in which additional spatial context is helpful (Fig. 5b). However, this approach does not allow the semantic segmentation of morphology smaller than a single rendered view. Higher-resolution classification requires a dense analysis of the rendered views. Similar to the approach taken by Boulch et al.[32], we solved the resulting image-to-surface mapping problem by rendering the cell views with spatially subdivided unique colors[33], thereby creating an efficient reversible mapping between the 2D view space and the 3D surface (Fig. 6a, b). We then trained a -FCN-VGG[34,35] based model on

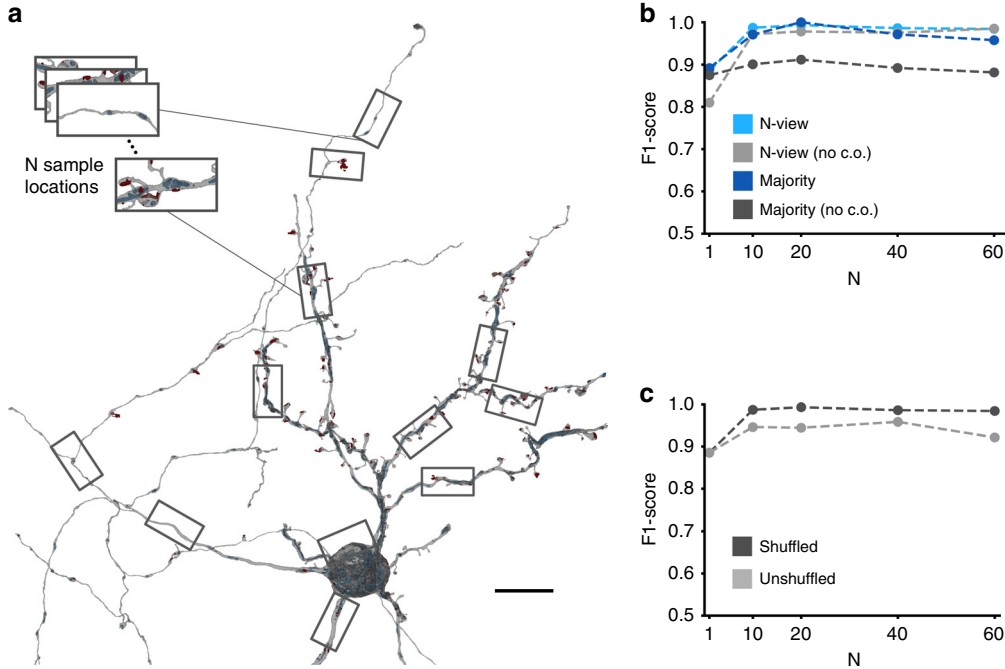

**Fig. 4** Cell-type inference using cellular morphology neural networks. **a** N-view fingerprint of a neuron reconstruction (red: synaptic junction; blue: mitochondria; green: vesicle cloud). **b** Class-weighted F1-scores of cell types with and without cell organelles (no c.o.). Light gray and light blue represent performances evaluated on individual N-views (class support for $N = 1$: excitatory axons (EA): 16,774, medium spiny neurons (MSN): 90,662, pallidal-like neurons (GP): 3958, interneurons (INT): 8806; $N = 10$: EA: 1658, MSN: 9048, GP: 394, INT: 880; $N = 20$: EA: 812, MSN: 4514, GP: 196, INT: 440; $N = 40$: EA: 388, MSN: 2245, GP: 97, INT: 220; $N = 60$: EA: 251, MSN 1489, GP: 65, INT: 146); dark tones represent performances on the SSV level after majority voting on multiple N-views (class support EA: 60, MSN: 39, GP: 3, INT: 2). **c** Class-weighted F1-score for shuffled (dark gray) and unshuffled (light gray) N-view stacks (class support as in **b**). Scale bar is 10 μm

the identification of dendritic spines (the training set contained five MSN reconstructions; 12.59 mm; 1.01 GV; 1652 μm³), a classification problem that was previously solved on manually traced skeletons, which made the automated identification of spines easy[11,36]. Automatically generated skeletons or surface meshes do not have the advantage that many skeleton endings are dendritic spines, which is likely a result of the implicit knowledge of human annotators. Instead of evaluating the classification performance on sampled locations, as done by us previously[11] (vertex-based evaluation in Supplementary Note 1), we tested the performance on a test set of 182 manually annotated synaptic contacts that were morphologically classified (see "Methods") as a spinous synapse ($N = 88$) or dendritic shaft synapse ($N = 94$) and obtained an F1-score of 0.978 (prec. 0.978, recall 0.978, F1-score spine head only 0.977; 2 views per location and $k = 20$). Interestingly, the classification performance did not improve further with more views (F1-score for $k = 20$ and 6 views: 0.978), likely because the PCA view alignment along the dendritic process already optimized the coverage. Note that the coverage saturates below 1.0 due to non-surface triangles (see Fig. 6c).

## Discussion

We demonstrated that CMNs are a versatile tool to analyze reconstruction fragments or entire cell reconstructions. Our approach adapts to the neurons' sparse coverage of the 3D volume by using distributed 2D multi-view projections instead of dense 3D models that end up processing many empty voxels[5,14,18,20]. While mainly developed for the analysis of neuron reconstructions, our proposed view-sampling method seems generally beneficial for the analysis of 3D objects that span large distances but still require high-resolution representations, as demonstrated by SnapNet which was developed independently for the semantic labeling of point clouds[32].

By combining the concept of CMNs with unsupervised triplet loss training[22], we created Neuron2vec embeddings that could serve as the basis for an unbiased morphological comparison of cells and cell types without requiring hand-designed features or allow neuron database queries using example neurites[16]. Additionally, the embedding can be used for visualization purposes, e.g., through coloring (Fig. 2), making morphologically different regions salient. The triplet-loss embedding could be compared to alternative unsupervised training paradigms (e.g., autoencoders[37,38] or generative query networks[39]), to evaluate whether the excellent supervised training results can be reached. CMNs showed improved performance in cell compartment classification in comparison to the previously used hand-designed features[11] and are likely more generalizable, since the morphology does not need to be parameterized first.

Based on the classification results of the glia CMN, we built a simple glia removal and merge error resolution algorithm, which could split neuron–glia mergers, but introduced additional false splits in about half of the newly created neurite components. However, these errors are considered to be easier to correct[8,40] and more importantly, their location is known, making guided proofreading possible. We would like to note that the other inferred cell types and neuronal compartments could be used in a similar manner or in combination as input to a more powerful graph cut segmentation algorithm, as proposed by Krasowski et al.[17].

Another application could be to directly evaluate the shape-plausibility of neurite fragments to detect errors[5,14] or to estimate the probability that separate SVs should be combined[41]. Applications requiring high-resolution segmentation (e.g., the localization of reconstruction errors) should especially benefit from the cell surface analysis that we used here for the classification of postsynaptic dendritic morphology with excellent performance

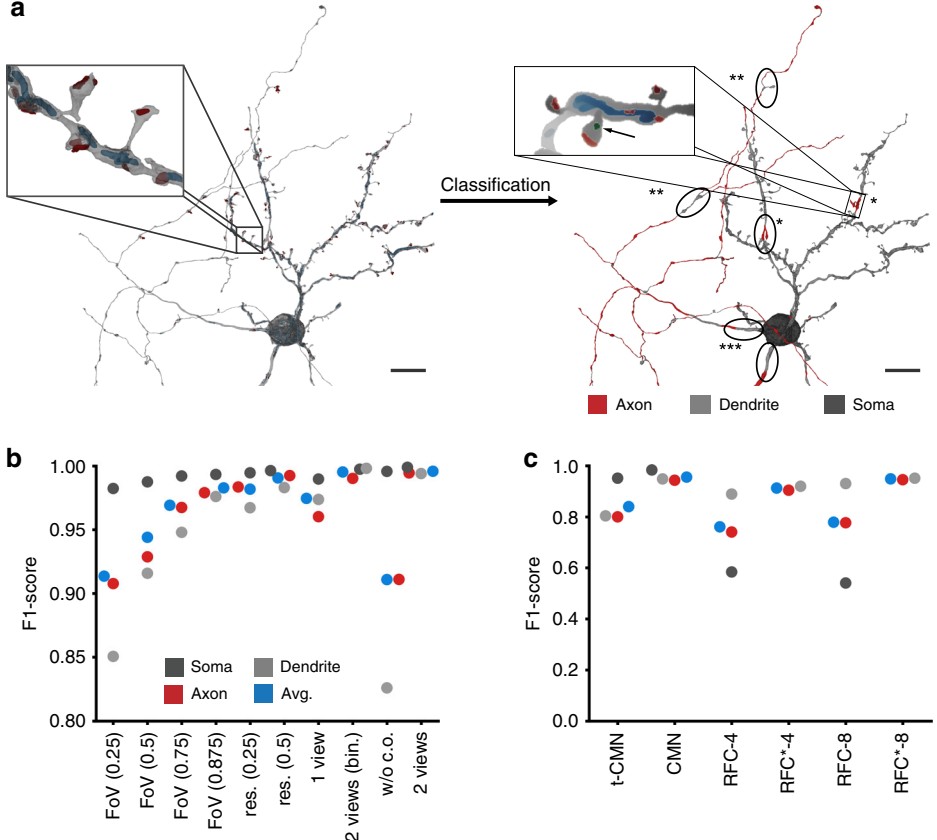

**Fig. 5** Cellular morphology neural network (CMN) prediction of subcellular compartments. **a** The 3909 rendering locations of the reconstruction were predicted as axon, dendrite, or soma. Local errors are indicated as follows: Single asterisks (*) indicate locations where vesicle clouds were falsely mapped to the cell (see inlay; red: synaptic junction, blue: mitochondria, green: vesicle cloud); Double asterisks (**) indicate branch points in axons where synaptic junctions occurred without vesicle clouds; Triple asterisk (***) indicates false dendrite predictions at proximal neurites. **b** Performance on the validation set (axon: 1078; dendrite: 1823; soma: 6139 multi-views) for different inputs. Left to right: Field of view (FoV) reduction of the two input views (with cell organelle channels) by image cropping (3/8 on each side); FoV reduction by cropping (1/4); FoV reduction by cropping (1/8); FoV reduction by cropping (1/16); resolution reduction by 4× downsampling; resolution reduction by 2× downsampling; single view perpendicular to the first and second p.c.; binarized input views; two views without cell organelle channels; two views at full resolution (256 × 128 pixels). **c** Comparison of skeleton and multi-view-based classifications measured on a skeleton node test set (color-code as in **b**) of 28 super-supervoxels ($N_{axon}$: 17,361; $N_{dendrite}$: 20,614; $N_{soma}$: 12,959). Asterisk (*) indicates that the RFC model was trained on binary label data (axon vs. dendrite) only. The CMN model was the two-view model evaluated in **b**; the triplet-CMN had $N_r = 9$, $d_z = 10$ and $k = 5$. Scale bars are 10 μm

(F1-score 0.98). PointNets[21] or PointCNNs[42], which can operate directly on mesh vertex data, might be an alternative, but their effectiveness for neuronal morphology classification remains to be demonstrated and compared to the projection-based CMNs.

## Methods

**EM data and used segmentation**. The analyzed EM data set (Area *X*, adult male zebra finch, >120 days post hatching) was acquired by J.K. through serial block-face scanning electron microscopy as reported previously[8,11] and had an extent of 96 × 98 × 114 μm³ with an *xyz*-resolution of 9 × 9 × 20 nm³ (total of 664 GV; 10,664 × 10,914 × 5701 voxel). The animal experiment was approved by the Regierungspräsidium Karlsruhe and performed in accordance with the laws of the German federal government. Meshes and skeletons were based on a FFN segmentation by M.J. and V.J.[8], including the over segmentation (all SVs) and the post-agglomeration SV graph (defines SSVs) combined with manual reconnects of orphan neurite fragments.

**Local scene rendering**. We used SV triangle meshes to efficiently render depth maps with PyOpenGL (http://pyopengl.sourceforge.net/) and EGL (for off screen rendering). For rendering, the model view matrix was rotated such that the first axis of the view was parallel to the main principal component of the object and clipped to an extent of 8 × 4 × 4 μm³. The rendering had a throughput of about 58 and 25 multi-views/s without and with cell organelles (see Supplementary Table 1) on a single CPU core with GPU acceleration, making it scalable in a cloud or high-performance computing environment.

PCA was applied to a subset (0.125) of the vertices within the clipping box that yielded the axes with highest variance (*x*, *y*, and *z*; decreasing variance) in good approximation. In this system, *xy* and *xz* represent the planes with the highest spatial variance, allowing alignment with the elongated neurite structure. By rotating around the *x* axis, orthographic depth-map projections of size 256 × 128 pixels were rendered and stored as unsigned 8-bit integer. For every SV, rendering locations were obtained by calculating its vertex density $\rho$ in a grid of voxels with size 2 × 2 × 2 μm³. In order to not oversample dense regions, the center coordinate of every voxel with $\rho > 0$ was used to calculate the mean of the vertices within a radius of 1 μm. The resulting set of coordinates, effectively a downsampled point cloud, was stored as the SVs rendering locations.

**Rendering cell organelles**. Three additional channels were generated next to the 2D depth-map views of the cell shape which contained the rendering of mapped cell organelles. Cell organelles (mitochondrion (MI); synaptic junction (SJ); vesicle cloud (VC)) with a relative overlap with SSVs above or within a certain threshold range (MI: >0.5, VC: >0.5, SJ: 0.2–0.8) were mapped to the SSV. Only objects with a minimum size (number of voxels) were considered during the overlap calculation (SJ: 498; VC: 1584; MI: 2786). Cellular organelle predictions were generated using 3D CNNs with ground truth volume details provided in Supplementary Table 4 of Dorkenwald et al., 2017[11] (see also Supplementary Note 2).

Meshes for the associated objects were extracted from a Gaussian-smoothed ($\sigma = 1$), distance-transformed (https://ukoethe.github.io/vigra/) binary 3D mask with marching cubes (contour value of 0; scikit-image http://scikit-image.org/).

The cell object meshes were rendered from the same perspective and resolution (256 × 128 pixels) as the corresponding SV views. The location's fingerprint finally consisted of the rotated views, each with four channels.

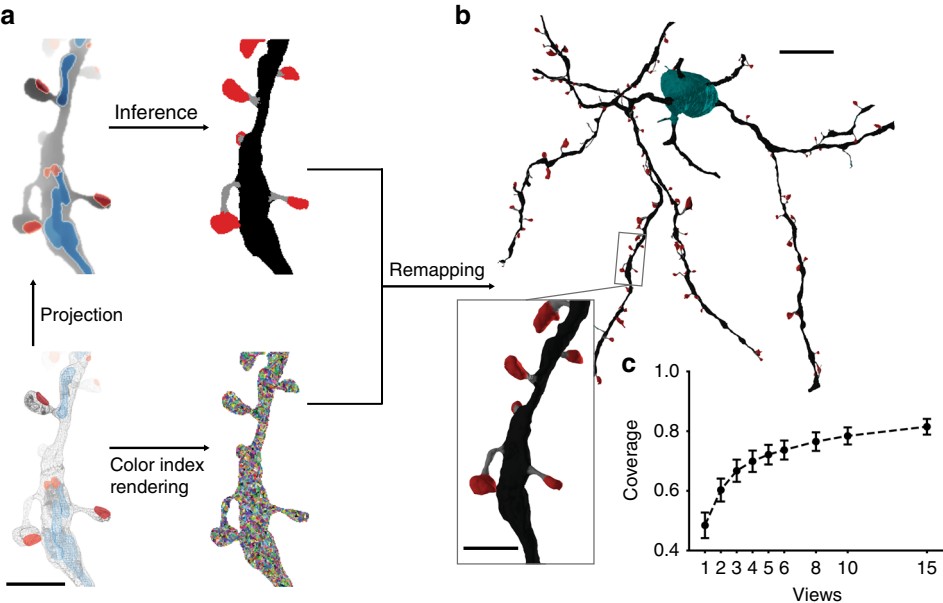

**Fig. 6** Sub-micron resolution semantic segmentation of cellular surfaces. **a** Bottom left: Wire-frame rendering of an flood-filling network-reconstructed mesh with mapped organelles. Top left: Depth map projection with organelle channels. Bottom right: Reversible two-dimensional-to-three-dimensional (3D) mapping based on unique color rendering. Top right: Dense prediction of semantic pixel labels (here spine head (red), neck (gray), dendritic shaft (black)) on rendered view. **b** Pixel labels are mapped back onto mesh faces through the unique colors, enabling a high-resolution 3D surface classification. **c** Mean ratio of triangle faces covered by all reconstruction multi-views depending on the number of views rendered per location ($N = 4$ cell reconstructions; error bars: s.d.). Scale bars are 2 μm in **a**, 10 μm in **b**, and 2 μm in the inset

**Automatic skeletonization of cell reconstructions**. The skeletons of the SV (provided by M.J. and V.J. and created using the TEASAR[43] algorithm) belonging to an SSV were combined iteratively. Edges between the spatially closest pair of nodes of all connected components were repeatedly added until only a single connected component remained.

We decreased the average edge length in the skeleton representations to approximately 150 nm by removing skeleton nodes (ignoring branch and end nodes). Nodes of degree 2 were removed and replaced by a single edge if the summed length $\lambda$ of the adjacent edges was below a threshold ($\lambda = 50$ nm) or if the dot product of their edges was >0.8 in combination with $\lambda = 500$ nm.

At every node, the cell radius was estimated by the median of the distance to the ten nearest vertices of the mesh. Total path lengths were calculated as the sum of all edges.

**Multi-view models for type classification**. Multi-views were sampled from the joint set of SV meshes of an entire SSV, and renderings generated at locations as described above.

To overcome RAM limitations, large SSVs (>$10^4$ SVs) were processed as subgraphs, defined by a breadth-first-search (extending 40 nodes) on the SV graph and starting at each SV. Multi-views were generated from the joint meshes of the 40 SVs at the sampled locations of the source SV.

The multi-view CNNs used seven (valid-mode) convolutional layers (number of filters, filter size, max-pooling size), each followed by a max-pooling layer, three fully connected layers, and a soft-max layer. For the glia prediction, e.g., the following architecture was used:

conv. L1: (13, (1, 5, 5), (1, 2, 2))
conv. L2: (17, (1, 5, 5), (1, 2, 2))
conv. L3: (21, (1, 4, 4), (1, 2, 2))
conv. L4: (25, (1, 4, 4), (1, 2, 2))
conv. L5: (29, (1, 2, 2), (1, 2, 2))
conv. L6: (30, (1, 1, 1), (1, 2, 2))
conv. L7: (31, (1, 1, 1), (1, 1, 1))
f.c. L1: 50 neurons
f.c. L2: 30 neurons
f.c. L3 + softmax: 2 neurons

The input to a model contained multi-views of shape (($b$, 20), ($f$, $c$), ($z$, 2), ($x$, 256), ($y$, 128)) with either one or four channels ($c$), batch size $b$, initial filter or channels $f$, and spatial axes $z$, $x$, and $y$. Note the auxiliary $z$ axis that was introduced to share filters for all views ($z$-filter size of 1). Number of filters varied slightly for different tasks. The models were trained using backpropagation with mini-batches (samples were drawn uniformly). During training, the ordering of the two views was inverted with probability $p = 0.5$.

If not stated otherwise, hyper-parameters were chosen to be:

batch size: 20
dropout rate[44] : 0.1
activation function: ReLu
Adam optimizer[45]
learning rate: $1 \times 10^{-4}$
momentum $\beta_{1,2}$: 0.9, 0.99
weight decay: $5 \times 10^{-4}$

A model with two output classes and one input channel had 35,770 trainable parameters and a computational cost of 2.4 Giga Ops. In all, 350,000 training iterations (with batch size of 20) on a GeForce GTX 980Ti took 20.1 h, which was about 0.0103 s per sample. Inference of 2754 rendering locations took 45.04 s, approximately 0.016 s per sample (see Supplementary Table 1).

**Neuron2vec embedding**. The architecture of the CMN encoder was used to learn a projection from the single view space $\mathbb{R}^{256 \times 128 \times 4}$ to a lower-dimensional latent space (embedding) $\mathbb{R}^d$ ($d$ dimensions) based on the triplet loss described in ref. [23]. Its architecture was identical to the one described in "Multi-view models for type classification," whereas max pooling was removed from conv. L2 and L6, dropout was restricted to conv. L3–L5, the f.c. L2 and the softmax from layer f.c. L3 were removed. The objective function was to keep the distance of the reference $x_{\text{ref}}$ to the similar input $x_+$ below the distance of $x_{\text{ref}}$ to the dissimilar input $x_-$. For the two similar views $x_{\text{ref}}$ and $x_+$, we used the two views of the same rendering location (rotated by $\varphi = 50°$), while the dissimilar view $x_-$ was sampled randomly from a different SSV. The clipping volume was set to $8 \times 4 \times 8$ μm$^3$. In order to take strong similarities of adjacent rendering locations into account, the view-sampling during training was adapted to sample similar views also from close-by rendering locations ($k$-nearest-neighbors with $k = 2$ and $k = 8$) instead of only using the rotated view at the same rendering location (Supplementary Fig. 2).

The loss was defined as $L_{\alpha>0} = \alpha + \lambda_2$ and $L_{\alpha\leq0} = \lambda_2$, with $\alpha = r_+ - r_- + \lambda_1$, $r_+ = \left|\tilde{x}_{\text{ref}} - \tilde{x}_+\right|_2$, $r_- = \left|\tilde{x}_{\text{ref}} - \tilde{x}_-\right|_2$, and $\lambda_1 = 0.2$ being a parameter to control the margin between data points. $\tilde{x} \subset \mathbb{R}^d$ represents the latent space of the triplet net. The second regularization term $\lambda_2$ was the mean norm of the reference, similar, and dissimilar view and acted as a counterpart to $\lambda_1$ by restricting the latent vectors to be small: $\lambda_2 = 1/3 \left( \left|\tilde{x}_{\text{ref}}\right|_2 + \left|\tilde{x}_+\right|_2 + \left|\tilde{x}_-\right|_2 \right)$. Our implementation was inspired by the one from A. Veit (https://github.com/andreasveit/triplet-network-pytorch).

The PCA was performed on the triplet network latent vectors of the 372 cell or cell fragment reconstructions, which were also used for querying the view-triplets during training. By taking only the first three principal components, every multi-view location was assigned an RGB value. The mesh was colored according to the nearest view color of every vertex.

**Glia classification**. For the glia classification model, only the depth maps of the SV were given as input. The training set contained 88,022 multi-views ($N_{neuron}$: 69,068; $N_{glia}$: 18,954) and the validation set 9695 ($N_{neuron}$: 7588; $N_{glia}$: 2107). Neuron views from the subcellular compartment ground truth (see section "Subcellular compartment classification") were extended by two additional axon reconstructions and used as samples for the negative class (31.29 mm 3.08 GV; 4989 $\mu m^3$). Glia views were generated from 118 manually annotated glia SV (path length: 337.45 mm; 13.40 GV; 21,706 $\mu m^3$). The performance was calculated based on multi-views and measured as F1-score. The classification threshold $\theta$ was set to the optimal F1-score on the validation set. The best model was retrained on training and validation set and applied to the whole data set. To remove background structures not connected to the central object of interest, connected component analysis was performed on the 2D multi-view images, followed by masking of the unconnected pixels.

The SVs were then classified by calculating the mean of all its multi-view predictions and thresholding with $\theta$. In addition, at least 70% of all multi-view predictions of a SV had to be glia for the assignment of this label.

Classification performance was measured by manually annotating 169 SV ($N_{glial}$: 85; $N_{neuron}$: 84; $N_{BBD < 8 \mu m}$: 57; $N_{BBD \geq 8 \mu m}$: 112). These SVs were sampled from 20,000 randomly drawn SSVs (training and validation samples were excluded), weighted by the number of views per SV. Only SSVs within the segmentation data set bounding box [470, 730, 30]–[10200, 10200, 5670] were considered.

**Top–down glia splitting**. In order to split glia fragments from neurons, a connected component analysis was applied to the SV graph of every SSV. Glia and neuron connected components (CCs) in the SV graph were computed on subgraphs induced by the classification (glia or neuron) of each SV. The CC size was estimated by calculating the BBD of its combined SVs. Glia CCs with a BBD ≥ 8.0 $\mu m$ were separated from the SV graph first. The remaining, small glia CCs (BBD < 8.0 $\mu m$) were assigned to the neuron class and the BBD was re-evaluated. Neuron CCs with a BBD < 8 $\mu m$ were removed and added to the glia graph. The purpose of this was to bridge small false glia/neuron predictions and thereby avoid false neuron splits.

The resulting class labels after the splitting procedure were evaluated on 12 randomly drawn SSVs with at least 1 introduced split. SVs were sorted by volume and manually labeled by P.J.S. until a major fraction of the reconstruction volume was covered. The average inspected volume coverage was 0.905 (proportion of inspected SVs weighted by their volume).

To evaluate the splitting performance, 181 SSVs were manually inspected by P.J.S. and J.K. The total number of neuron components and the number of neuron components that were not split into several parts were identified, i.e., those in which all splits preserved the neuron as a single component. Removing, for example, a small fragment from an SSV (Supplementary Fig. 6d) was not considered a split and the SSV therefore counted as a correct component. In contrast, an axon passing through a falsely merged glia and being split into two components was not added to the number of correct components, but its two components were added to the total number of resulting neuron components. Note that we also counted a component as incorrect in case the removed fragment (predicted as glia) only virtually disconnected the neurite (Supplementary Fig. 6e), e.g., at the data set boundary, when it could be reasonably assumed that it would continue in a larger data set.

For the evaluation of the large SSV, which was split into 593 SSV (516 neuron and 77 glia components predicted), we inspected a random subset of 50 neuron components.

**Reconstructing glial cells**. SV graph edges were added between the sample locations of the collection of all splitted glia SV by identifying the $k$-nearest neighbors ($k = 15$, maximum distance: 10 $\mu m$; weighted by Euclidean distance).

Twenty-seven somata of putative astrocytes were identified in the data set and every glia SV was assigned to its closest soma (shortest path using Dijkstra's algorithm).

**Bloodvessel prediction**. The input data ($zxy$ ordering) for the bloodvessel CNN was downsampled by a factor of eight in all dimensions. A cube of size (256, 437, 287) was densely labeled using KNOSSOS to obtain training data. The used network had the following architecture:

```
conv. L1: (24, (1, 6, 6), (1, 2, 2))
conv. L2: (27, (1, 5, 5), (1, 2, 2))
conv. L3: (30, (1, 5, 5), (1, 1, 1))
conv. L4: (33, (1, 4, 4), (2, 1, 1))
conv. L5: (36, (3, 4, 4), (1, 1, 1))
conv. L6: (39, (3, 4, 4), (1, 1, 1))
conv. L7: (42, (2, 4, 4), (1, 1, 1))
conv. L8: (45, (1, 4, 4), (1, 1, 1))
conv. L9: (48, (1, 4, 4), (1, 1, 1))
conv. L10: (48, (1, 1, 1), (1, 1, 1))
conv. L11 + softmax: (2, (1, 1, 1), (1, 1, 1))
```

The dense predictions of the data set were thresholded at 0.98. Meshes were created as described above ("Rendering cell organelles").

**Neuron-type classification**. The 2-views generated at each rendering location were re-used to construct a set of $N$-views for every SSV by the following procedure: The collection of all $M$ 2-views of an SSV was split into $2M/N$ random sets (drawn without replacement) each of size $N$. If $2M < N$, the set was filled by randomly drawing from the existing views.

The model architecture was identical to the model used for glia classification, except for a reduced batch size, a dropout rate of 0.08, and a learning rate schedule defined as exponential decay, with decay rate of 0.98 per 1000 steps. The input shape was (1, 4, $N$, 256, 128).

We used 402 manually traced (skeletonized) cells to identify their corresponding SSV, which were split into training set ($N_{train}$: 301; $N_{EA}$: 177, $N_{MSN}$: 114, $N_{GP}$: 6, $N_{INT}$: 6) and test set ($N_{test}$: 101; $N_{EA}$: 60, $N_{MSN}$: 39, $N_{GP}$: 3, $N_{INT}$: 2) with labels, that corresponded to the broad biological classes found in the data set (EA, MSN, GP, INT).

During batch creation while training, the $N$-views were generated by randomly drawing from the corresponding SSV views. Every batch contained an equal number of SSVs for each class. The classification was performed using argmax on the output of the softmax layer and the majority vote of the corresponding $N$-view classifications was used for SSV classification.

The support-weighted average F1-score of all classes was evaluated on $N$-views and on a SSV level after majority vote (carried out on all its $N$-view predictions).

**Subcellular compartment classification**. The cellular compartments of 33 neurites were manually annotated and axon, dendrite, and soma views were generated, which were split into a training set ($N_{train}$: 80,370 views; $N_{dendrite}$: 10,004; $N_{axon}$: 41,424; $N_{soma}$: 28,942) and a validation set ($N_{validation}$: 9,040; $N_{dendrite}$: 1,078; $N_{axon}$: 1,823; $N_{soma}$: 6139). During training, we applied class weights for loss computation to address imbalances in their frequency (dendrite: 2, axon: 1, soma: 1). Performance was measured with the F1-score of the multi-view classification using argmax on the softmax output. The best model was again retrained on the whole ground truth data for the data set prediction.

Classification of so far unclassified locations within neurons was performed by assigning it the label of the closest classified location (Voronoi partitioning with Euclidean distance).

All SSV skeleton nodes were manually labeled by P.J.S. using KNOSSOS. In order to enable a direct comparison between the RFC and CMN model, the skeleton-node locations were used for the extraction of the hand-designed features. CMN and kNN predictions were mapped to the skeleton nodes using the nearest neighbors on the multi-view locations. As in Dorkenwald et al.[11], hand-designed features were computed for every skeleton node (context of 4 $\mu m$ and 8 $\mu m$ maximal traversed path length from the source node). Only properties of nodes visited during the traversal were considered for the source node statistics.

A total of 23 features were extracted from the collected properties at each node: mean and standard deviation (s.d.) and histogram (10 bins) of the encountered node diameters, mean of node degrees, node density within a box with edge length of 2-times the context-range (either 4 or 8 $\mu m$) number of cell organelles and mean and s.d. of their size for mitochondria, synaptic junctions, vesicle clouds. The RFC was trained on the same training data as the CMN to classify each node as axon, dendrite, or soma using argmax on the resulting class probability.

The sliding window majority vote was performed on the cell reconstruction skeletons. Every skeleton node was assigned the majority label found in a set of adjacent node/multi-view predictions, which were collected within a maximum of 12.5 $\mu m$ traversal length along the skeleton.

**High-resolution semantic segmentation of surfaces**. The training data were generated by rendering multi-views (5 different perspectives) from the rendering locations of 5 reconstructions (training: 24,248 views; validation: 6062 views) with label-dependent vertex colors. Skeleton nodes were manually annotated as neck, head, shaft, or soma/axon, which were then mapped to the mesh vertices with Voronoi partitioning (Euclidean distance). To smooth label boundaries, each vertex was assigned the majority label of 40 vertices found by a breadth-first search on the vertex graph of the reconstruction. Graph edges were added between vertices with a distance of up to 120 nm. Only rendering locations with annotated skeleton nodes within 2 $\mu m$ were considered.

We used a FCN-VGG architecture[35] with 13 layers (adopted from https://github.com/pochih/FCN-pytorch by P.-C. Huang) to perform pixel-wise multi-class (neck, head, shaft, background, and axon/soma) classification on single views with four channels (cell, mitochondria, synaptic junctions, and vesicle cloud shapes). It was trained using backpropagation with mini-batches (images were flipped in $x$ or $y$ with probability $P = 0.5$), Adam optimizer ($\beta_{1,2}$: 0.9, 0.999; weight decay: $5 \times 10^{-4}$), initial learning rate of $4 \times 10^{-3}$ (exponential decay with 0.99), and Lovász-Softmax loss[46].

To assign pixel labels back to the mesh vertices, an additional view of the faces was rendered using color buffering with a unique ID per face. This allowed to perform a majority vote of all collected labels corresponding to a single vertex as classification. Subsequently, we applied a kNN classification to propagate predicted labels to vertices, which were not covered by the rendered color map.

The synapse test set was generated as a random set of head and shaft synapses collected from four different reconstructions, which were also used to calculate the cell surface fraction captured by the multi-views. The dendritic tree for the

per-vertex evaluation was manually annotated on skeleton node level and then propagated to the mesh vertices as described above for the GT generation.

**Computing infrastructure**. The used parallel computing environment consisted of 18 nodes, each equipped with 20 cores (Intel Xeon CPU E5-2660 v3 @ 2.60GHz), 2 GeForce GTX 980Ti, and 256 GB of RAM. Compute jobs were managed using SGE QSUB or SLURM.

## Data availability

The data sets generated and analyzed during the current study are available from the corresponding author on reasonable request.

## Code availability

The used code, network architectures, classes for handling the inference, processing, and storage of the segmentation data can be found in the open source SyConn repository on GitHub (https://github.com/StructuralNeurobiologyLab/SyConn/).

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

## Acknowledgements

We thank Winfried Denk for enabling this work in his department and many helpful discussions; Michale S Fee, Joseph R Scherrer, and Maria Kormacheva for helpful

comments; the KNOSSOS team for support; Rangoli Saxena, Mariana Shumliakivska, and Atul T Mohite for code contributions; Marius Killinger and Martin Drawitsch for help with the ELEKTRONN library; and Julia Rogowska and Mariana Shumliakivska for proofreading tracings and ground truth generation. We also thank Christian Guggenberger and his team at the MPCDF facility in Garching, Germany for compute cluster support.

## Author contributions

P.J.S. and J.K. performed and designed experiments and wrote the manuscript. M.J. and V.J. contributed the data set FFN segmentation and wrote the manuscript. SD contributed code for the experiments.

## Additional information

**Competing interests:** J.K. holds shares of ariadne-service gmbh. The remaining authors declare no competing interests.

