## [Peer Review File · Nature Communications]

Reviewers' comments:

Reviewer #1 (Remarks to the Author):

This work demonstrates that neural networks trained on randomly sampled 2D projections of cell fragments are capable of:

- (1) discriminating between glia and neuron cell fragments,
- (2) identifying subcellular components (soma, axon, dendrite), and
- (3) identifying neuron types.

The main novelty compared to prior art is the application of 2D multi-view projections (Su et al., cited) to fragments of automatically reconstructed cells and some of their components. The differences to previous work for each of the claims above are:

- (1) Rolnick et al. (cited) used 3D CNNs to find likely merge errors, but did not classify glia explicitly. The method presented here does additionally identify and thus reconstruct glia cells.
- (2) Krasowski et al. (cited) also use local cues to identify axons and dendrites, but embed those in a global joint segmentation and component identification problem. The method presented here assumes that a segmentation is already available.
- (3) Previous work by the authors (cited) shows that cell types can be inferred from local skeleton features using an RFC. The method presented here does not require a skeletonization.

Although only evaluated on a single zebra finch dataset, the presented method is likely of interest to other researchers. The identification of glia cells, subcellular components, or neuron types would aid reconstruction and interpretation of the results in other model organisms or imaging techniques. Whether this method will be widely adapted by others on different datasets is currently hard to tell. The presented method assumes that a segmentation of cells and their organelles (mitochondria, synaptic junctions, and vesicle clouds) is available. To gauge the costs of applying this method to a different dataset, the required labeling effort and prediction times for cells and their organelles should be reported as well.

The claims made by the authors are generally backed up by experiments on the investigated dataset, except for the error correction capabilities and the improvements on axon vs. dendrite classification in comparison to prior methods (see details below). The paper would be strengthened further if it could be shown whether the findings generalize to other model organisms or imaging techniques. However, such an evaluation requires substantial effort, which is likely beyond what would be reasonable to ask for in a revision.

The method description is in most parts detailed and generally provides enough information to reproduce the method, with two exceptions being the segmentation of cell organelles and the description of the splitting procedure for error correction (see details below).

Glia detection and top-down segmentation error correction

It seems the splitting performance in the top-down segmentation error correction is reported in terms of F1-scores for the classification of glia SVs. To back up the claim that the glia classifier can be used for error correction, the number of false merges and false splits before and after the splitting procedure should be reported.

The description of the splitting procedure (page 6, third paragraph) is largely unclear. Is the splitting only done on glia components with BBD $\geq 8\mu\text{m}$? What is a "predicted glia merger"? What are the "unit weights", and how are they different from "volume weights"? The legend of corresponding Fig. S5 refers to S4e, which doesn't exist.

Cellular compartment identification

The conclusion that the CMN outperforms RFCs on axon vs. dendrite classification is not sufficiently backed up: RFC*-8 seems to be par with CMN.

Neuron type classification

It is not clear from the caption of Fig. 4 why the class support for majority voting is significantly smaller than for the N-views results.

High-resolution semantic segmentation of neurite surfaces

Results are reported in terms of F-scores of identified spinous and dendritic shaft synapses. It should be mentioned how the classification was obtained from the remapped predictions.

Minor points

- Figures in S3, 5b: Sort x axis, use bar plots; or consider a table instead
- Disambiguate different F1-scores: On page 6, first paragraph, SVs with a BBD $\geq 8\mu\text{m}$ are reported to have a better F1-score than others (0.964 vs 0.868); however, the overall F1-score is 0.979. It seems different scores are considered (glia vs. non-glia; non-glia vs. glia; average), this should be made more obvious.
- Page 6, third paragraph: "ECS" is not introduced.
- Page 8, last paragraph: "that lower" -> "that at lower"
- Page 10, first paragraph: "Fig 6c" -> "Fig 5c"
- Caption of Fig 5 says * indicates that the RFC was trained on binary label data, text (page 10, top) says * indicates RFC models tested on entire axons and dendrites.
- Page 11, second paragraph: refers to Fig. 6d,e, which don't exist.

Reviewer #2 (Remarks to the Author):

This paper introduces a convolutional neural network with multiview 2D input for various cellular morphology-related analysis tasks. First, the paper describes the neural network model and implementation details on generating multiview input. Then the paper walks through four different applications: unsupervised neuron2vec embedding, supervised glia detection, supervised neuron type classification and supervised cellular compartment identification.

Strengths

- application novelty: the applications in the paper are novel and interesting. Especially the neuron2vec embedding is useful for unbiased process analysis, and the glia removal is helpful for better neurite reconstruction
- model simplicity: the paper uses one backbone model for all applications with a slight change of the input or loss function.
- solid experiment: the paper provides extensive results for each application with careful experiment design.
- detailed/clear explanation: the paper is easy to read with a detailed explanation for each computation step. The figures are well-made with self-sufficient captions.

Weaknesses

- lack of an overview figure. The CMN model is applied to various applications with different training strategies. It would be much clearer if the paper had one overview figure to describe all applications to help readers better understand the similarities and differences between them.
- the results section is currently not ordered in an intuitive way and could be improved. For example:
 - = local input: each sample location
 - # semantic classification (glia-neurite, axon-dendrite-soma)
 - # general purpose feature
 - = global input: entire neuron
 - # for type classification
- model novelty: the main model is a standard multiview CNN model used in previous work [18,19,32]. Besides the input generation and scale selection, there seems to be little novelty.

Reviewers' comments, response in *italic*:

Reviewer #1 (Remarks to the Author):

This work demonstrates that neural networks trained on randomly sampled 2D projections of cell fragments are capable of:

- (1) discriminating between glia and neuron cell fragments,
- (2) identifying subcellular components (soma, axon, dendrite), and
- (3) identifying neuron types.

The main novelty compared to prior art is the application of 2D multi-view projections (Su et al., cited) to fragments of automatically reconstructed cells and some of their components. The differences to previous work for each of the claims above are:

- (1) Rolnick et al. (cited) used 3D CNNs to find likely merge errors, but did not classify glia explicitly. The method presented here does additionally identify and thus reconstruct glia cells.
- (2) Krasowski et al. (cited) also use local cues to identify axons and dendrites, but embed those in a global joint segmentation and component identification problem. The method presented here assumes that a segmentation is already available.
- (3) Previous work by the authors (cited) shows that cell types can be inferred from local skeleton features using an RFC. The method presented here does not require a skeletonization.

Although only evaluated on a single zebra finch dataset, the presented method is likely of interest to other researchers. The identification of glia cells, subcellular components, or neuron types would aid reconstruction and interpretation of the results in other model organisms or imaging techniques. Whether this method will be widely adapted by others on different datasets is currently hard to tell. The presented method assumes that a segmentation of cells and their organelles (mitochondria, synaptic junctions, and vesicle clouds) is available. To gauge the costs of applying this method to a different dataset, the required labeling effort and prediction times for cells and their organelles should be reported as well.

Response: We agree with the reviewer that it remains unclear how widely adopted our method will be in the end, but the approach seems to be hard to beat in classification performance given the already very good classification results (e.g., F1 score of 0.955 for compartment classification). We added now several numbers to the manuscript that should help the reader to gauge the cost of applying CMNs (updated Supplementary Table 1) and estimates of the labeling effort (see new Supplementary Note 2).

The claims made by the authors are generally backed up by experiments on the investigated dataset, except for the error correction capabilities and the improvements on axon vs. dendrite classification

in comparison to prior methods (see details below). The paper would be strengthened further if it could be shown whether the findings generalize to other model organisms or imaging techniques. However, such an evaluation requires substantial effort, which is likely beyond what would be reasonable to ask for in a revision.

Response: We fully agree that evaluations on other model organisms or with different imaging techniques would strengthen the manuscript, but do not have the resources to perform these at the moment. While it remains unclear whether our method will work as well on e.g., insect neuron classification, due to their compact and sometimes extraordinary morphology, we would find it surprising if data from other vertebrates, including mammals, would lead to very different results, due to the overall very similar basic morphology of neurons (e.g., spine shapes and structuring into axon, dendrite and soma).

The method description is in most parts detailed and generally provides enough information to reproduce the method, with two exceptions being the segmentation of cell organelles and the description of the splitting procedure for error correction (see details below).

Response: The segmentation of the cell organelles was described in detail in Dorkenwald et al., 2017 and we added a reference in the Methods (see section "Rendering cell organelles"). We acknowledge the lack of detail in regard to the splitting procedure and have attempted to resolve this issue (see below).

Glia detection and top-down segmentation error correction

It seems the splitting performance in the top-down segmentation error correction is reported in terms of F1-scores for the classification of glia SVs. To back up the claim that the glia classifier can be used for error correction, the number of false merges and false splits before and after the splitting procedure should be reported.

Response: We would like to thank the reviewer for spotting this and have performed new evaluation experiments to further back up our claims. In addition to the F1-score, which we still consider important because it indicates the overall correctly classified volume fraction, we inspected all modifications (n=181 connected components) that were introduced to the dataset-wide agglomeration graph by the glia removal procedure. At the (in our opinion acceptable) cost of introducing additional splits into about half the affected neurons (considered less harmful in the context of automatic neuron reconstruction, see e.g., Januszewski et al., 2018), we could resolve false mergers in 139 neuron-glia connected components, some of which had agglomerated more than 10 neuron fragments into a single huge connected component, making manual proofreading very difficult. The results section and methods were updated in this regard.

The description of the splitting procedure (page 6, third paragraph) is largely unclear. Is the splitting only done on glia components with BBD $\geq 8\mu\text{m}$? What is a "predicted glia merger"? What are the "unit weights", and how are they different from "volume weights"? The legend of corresponding Fig. S5 refers to S4e, which doesn't exist.

Response: See above, we have rewritten large parts of these descriptions and also revised the corresponding methods section to improve understandability.

Cellular compartment identification

The conclusion that the CMN outperforms RFCs on axon vs. dendrite classification is not sufficiently backed up: RFC*-8 seems to be par with CMN.

Response: This was indeed insufficiently described so far, but we think that the conclusion is in fact sufficiently backed up. As shown in Fig. 5 and S8 and now explained in greater detail in the main text, the RFC models were applied to a narrower classification problem. While it is conceivable that the RFC soma classification performance could be improved as well with additional software engineering and feature design effort to recognize the peculiarities of the automatic neurite reconstruction in this regard (see suppl. Fig S8a), we wanted to compare the CMN approach to an existing and established RFC method (Dorkenwald et al., 2017), for a direct comparison on the same data.*

Neuron type classification

It is not clear from the caption of Fig. 4 why the class support for majority voting is significantly smaller than for the N-views results.

Response: The class support is reduced through majority voting, since the evaluation takes place on the global neurite level. We made this now more explicit in the figure caption.

High-resolution semantic segmentation of neurite surfaces

Results are reported in terms of F-scores of identified spinous and dendritic shaft synapses. It should be mentioned how the classification was obtained from the remapped predictions.

Response: We clarified this in the manuscript.

Minor points

- Figures in S3, 5b: Sort x axis, use bar plots; or consider a table instead

Response: Resolved.

- Disambiguate different F1-scores: On page 6, first paragraph, SVs with a BBD $\geq 8\mu\text{m}$ are reported to have a better F1-score than others (0.964 vs 0.868); however, the overall F1-score is 0.979. It seems different scores are considered (glia vs. non-glia; non-glia vs. glia; average), this should be made more obvious.

Response: Resolved.

- Page 6, third paragraph: "ECS" is not introduced.

Response: Resolved.

- Page 8, last paragraph: "that lower" -> "that at lower"

Response: Resolved.

- Page 10, first paragraph: "Fig 6c" -> "Fig 5c"

Response: Resolved.

- Caption of Fig 5 says * indicates that the RFC was trained on binary label data, text (page 10, top) says * indicates RFC models tested on entire axons and dendrites.

Response: Resolved.

- Page 11, second paragraph: refers to Fig. 6d,e, which don't exist.

Response: Resolved.

Reviewer #2 (Remarks to the Author):

This paper introduces a convolutional neural network with multiview 2D input for various cellular morphology-related analysis tasks. First, the paper describes the neural network model and implementation details on generating multiview input. Then the paper walks through four different applications: unsupervised neuron2vec embedding, supervised glia detection, supervised neuron type classification and supervised cellular compartment identification.

Strengths

- application novelty: the applications in the paper are novel and interesting. Especially the neuron2vec embedding is useful for unbiased process analysis, and the glia removal is helpful for better neurite reconstruction
- model simplicity: the paper uses one backbone model for all applications with a slight change of the input or loss function.
- solid experiment: the paper provides extensive results for each application with careful experiment design.
- detailed/clear explanation: the paper is easy to read with a detailed explanation for each computation step. The figures are well-made with self-sufficient captions.

Response: We would like to thank the reviewer for the positive feedback.

Weaknesses

- lack of an overview figure. The CMN model is applied to various applications with different training strategies. It would be much clearer if the paper had one overview figure to describe all applications to help readers better understand the similarities and differences between them.

Response: This was a very good suggestion, and we incorporated now a panel into Figure 1 that visualizes and compares the different applications that were tested.

- the results section is currently not ordered in an intuitive way and could be improved. For example:

= local input: each sample location

semantic classification (glia-neurite, axon-dendrite-soma)

general purpose feature

= global input: entire neuron

for type classification

Response: We think that the current ordering is in fact intuitive as well, starting with the general concept, followed by an unsupervised application (Neuron2vec) and the different supervised applications in the end. This should now have become more clear with the new overview panel of Figure 1.

- model novelty: the main model is a standard multiview CNN model used in previous work [18,19,32]. Besides the input generation and scale selection, there seems to be little novelty.

Response: We agree that the novelty is more in the input generation and usage of the models, and not so much in CNN architecture design, but actually do not consider this to be a weakness. The high pace at which new and improved CNN architectures for general 2D classification / semantic segmentation are presented means that all current models will likely be outdated at the end of the year. The application itself on the other hand, i.e. the automatic morphology analysis of high-resolution cell reconstructions in the way proposed, can be expected to increase in usage, with the recent speed-ups in volume electron microscopy and improvements in segmentation methods. While it is plausible that models that can directly operate on 3D mesh representations (e.g., PointNet, Qi et al., 2017; <https://arxiv.org/abs/1612.00593> or similar) are as powerful, these will also require an efficient input data partitioning scheme, which is part of the pipeline we propose.

Reviewers' comments:

Reviewer #1 (Remarks to the Author):

The description of the splitting procedure is more clear now. However, the evaluation of the top-down segmentation error correction still raises questions.

The text mentions that 882 splits have been introduced to 181 neurites* (SSVs in the following). This would result in 1063 new SSVs (not mentioned in the text). It is also said that in about half of the newly created SSVs (which would be around 584 SSVs, not mentioned in the text) "a new topological split" was introduced. It is not entirely clear what that means. On one end of the spectrum, if all those 584 SSVs would belong to the same original SSV, it would mean that 583 incorrect splits have been introduced (and need to be manually fixed). On the other end, if all those 584 SSVs would split 292 original SSVs into two parts, it would mean that 292 incorrect splits have to be fixed. The latter can obviously not be the case since only 181 original SSVs are used. This part needs clarification.

In particular, a straight answer to the following question would immediately inform the reader of the benefits of the proposed splitting procedure: How many wrong splits and merges would have to be fixed by hand before and after the splitting procedure?

* Since some of the 181 SSVs are reportedly glia only, they should not be called neurites here.

Minor points:

- Fig S6e: should be 10
- Fig S6 caption d: "left" and "right" swapped?
- What is the difference between plots in Fig S8b and Fig 5c?

Reviewer #2 (Remarks to the Author):

I recommend accepting the revised manuscript.

I agree with the authors that as an application paper, the main contribution is to adapt the available deep learning model for cellular morphology tasks for better performance instead of devising new architectures. Although the effectiveness of the model is only demonstrated on one EM dataset from zebra finch, the paper does contain useful ingredients for other imaging modality and species.

The revised manuscript is significantly improved. The overview figure makes the structure of the paper easier to follow. The additional evaluation results better show the significant improvement upon the prior art. The revised text fills in more details on experimental procedures.

Reviewers' comments:

Reviewer #1 (Remarks to the Author):

The description of the splitting procedure is more clear now. However, the evaluation of the top-down segmentation error correction still raises questions.

The text mentions that 882 splits have been introduced to 181 neurites* (SSVs in the following). This would result in 1063 new SSVs (not mentioned in the text). It is also said that in about half of the newly created SSVs (which would be around 584 SSVs, not mentioned in the text) "a new topological split" was introduced. It is not entirely clear what that means. On one end of the spectrum, if all those 584 SSVs would belong to the same original SSV, it would mean that 583 incorrect splits have been introduced (and need to be manually fixed). On the other end, if all those 584 SSVs would split 292 original SSVs into two parts, it would mean that 292 incorrect splits have to be fixed. The latter can obviously not be the case since only 181 original SSVs are used. This part needs clarification. In particular, a straight answer to the following question would immediately inform the reader of the benefits of the proposed splitting procedure: How many wrong splits and merges would have to be fixed by hand before and after the splitting procedure?

While a detailed and non-estimated answer to "How many wrong splits and merges would have to be fixed by hand before and after the splitting procedure [in the data set]?" would be ideal and is indeed of outstanding interest to the authors, it would require an evaluation which is out of scope of this manuscript (and it is still a matter of debate in the field which errors are deemed acceptable and which not).

To still address the concern raised (i.e. whether the splitting procedure is beneficial or not) we have refined the manuscript substantially and added more explanations to the methods and main text, scrutinized our evaluation procedure again and re-evaluated the effects of the splitting procedure to provide a clearer picture of its benefits and limitations (see also extended discussion).

We would also like to note that the underlying segmentation was already evaluated in Januszewski et al., 2018 (as pointed out now), which is why we restricted our analysis to the changes introduced by the new splitting procedure.

We consider the glia splitting procedure furthermore an application example of the developed morphology analysis methodology - our key contribution - and are aware, as written in the discussion, that more sophisticated methods such as optimized graph cuts (Krasowski et al., 2018) could potentially make better use of the gained morphological data.

* Since some of the 181 SSVs are reportedly glia only, they should not be called neurites here.

Correct, done.

Minor points:

- Fig S6e: should be 10

Done.

- Fig S6 caption d: "left" and "right" swapped?

Done.

- What is the difference between plots in Fig S8b and Fig 5c?

While Fig. 5c shows the class-wise and average F1-score of the different classification models evaluated on skeleton node level (see Methods), Supplementary Figure 8b reports those performance criteria after a sliding window majority voting along the skeleton (see Methods). See Fig S8c for a direct comparison. We added additional references to Fig. 5c and Fig S8b in the figure caption of Fig S8c to clarify this.

Reviewer #2 (Remarks to the Author):

I recommend accepting the revised manuscript.

I agree with the authors that as an application paper, the main contribution is to adapt the available deep learning model for cellular morphology tasks for better performance instead of devising new architectures. Although the effectiveness of the model is only demonstrated on one EM dataset from zebra finch, the paper does contain useful ingredients for other imaging modality and species.

The revised manuscript is significantly improved. The overview figure makes the structure of the paper easier to follow. The additional evaluation results better show the significant improvement upon the prior art. The revised text fills in more details on experimental procedures.

REVIEWERS' COMMENTS:

Reviewer #1 (Remarks to the Author):

The analysis of the proposed splitting procedure improved substantially. I suggest accepting the manuscript.

--Jan Funke